# Healing the Whole: An International Review of the Collaborative Care Model between Primary Care and Psychiatry

**DOI:** 10.3390/healthcare12161679

**Published:** 2024-08-22

**Authors:** Veronica Hernandez, Lucy Nasser, Candice Do, Wei-Chen Lee

**Affiliations:** 1John Sealy School of Medicine, University of Texas Medical Branch, Galveston, TX 77555, USA; velherna@utmb.edu (V.H.); ljfisher@utmb.edu (L.N.); cado@utmb.edu (C.D.); 2Department of Family Medicine, University of Texas Medical Branch, Galveston, TX 77555, USA

**Keywords:** collaborative care model, anxiety, depression, PTSD, cost-effective, sustainable

## Abstract

The collaborative care model (CCM) was created to improve the delivery of mental health care and is reported to improve access, enhance treatment outcomes, and reduce healthcare costs. To understand the impacts of the CCM on symptom management, diverse populations, and sustainability in healthcare systems, a systematic review was conducted. Several databases were searched for articles assessing the CCM. The inclusion criteria limited the studies to those (1) published between January 2008 and January 2024; (2) written in the English language; (3) analyzing adult patients; (4) analyzing symptom improvement in major depressive disorder, generalized anxiety disorder, or post-traumatic stress disorder; and (5) fitting the given definition of a CCM. We identified 9743 articles. Due to missing information or duplication, 4702 were excluded. The remaining articles were screened, yielding 468 articles for full-text analysis, of which 16 articles met the inclusion criteria. Of these articles, five primarily focused on individual patient outcomes, five focused on specific populations, and six reviewed system impacts; eleven articles studied US populations and five studied international populations. An analysis revealed that in 12 of the final articles, the CCM led to a statistically significant improvement in anxiety and depression symptoms with viable implementation and sustainability strategies. The CCM is an effective method for improving patient symptoms and can be potentially affordable in healthcare systems.

## 1. Introduction

Several collaborative models exist between primary and mental health care. In the co-located care model, primary care practitioners and mental health specialists (e.g., psychiatrists) are located in the same physical location so that patients may receive care for both physical and mental health during a single visit, yet both providers remain autonomous in delivering care [1]. Integrated care involves both sets of providers, primary care and mental health delivered by either a psychologist or psychiatrist, working as a team within the same electronic health record system [2]. The collaborative care model (CCM) was recently developed to improve on both previous models in an attempt to provide the highest degree of communication between primary and mental health providers [3]. The CCM has been successful in managing the mental health conditions commonly encountered in the primary care setting, including major depressive disorder (MDD), generalized anxiety disorder (GAD), and post-traumatic stress disorder (PTSD) [4,5,6,7].

The CCM was developed at the University of Washington as a systemic strategy to treat behavioral health conditions in primary care settings through the integration of care managers and psychiatric consultants [3]. The CCM includes three major components (Figure 1): a primary care provider (PCP), a licensed mental health provider, and a care manager. PCPs provide integrated accessible healthcare services and are accountable for addressing a large majority of personal healthcare needs, developing a sustained partnership with patients, and practicing in the context of family and community [3]. A licensed mental health provider, who provides the PCP with consultation and support, can be a psychiatrist, psychologist, master’s-level mental health clinician, or mental health nurse practitioner. Lastly, a care manager is a specially trained healthcare professional, oftentimes a specialist nurse, who coordinates care for patients with chronic diseases. When a licensed social worker acts as a care manager, they can also work with patients to address social risk factors, but this activity is not required for the CCM.

The CCM has gained international recognition as a promising approach to improving access to mental health care, enhancing treatment outcomes, and reducing healthcare costs [8,9]. Yet, limited research has been conducted evaluating the outcomes across various regions. With depression/MDD, anxiety/GAD, and PTSD among the most prevalent mental health conditions, it is crucial to evaluate what methods are effective in delivering care to patients worldwide [10,11]. Implementation can vary significantly across the world due to diverse patient demographics, economic factors, and governmental structures [12]. Demographic factors such as the age distribution, prevalence of mental health conditions, and cultural norms can influence the design and impact of CCM initiatives [12]. In addition, healthcare disparities in rural areas stemming from limited infrastructure and fewer providers require innovative approaches to implement the CCM. 

Healthcare financing systems, such as public and private insurance models and out-of-pocket expenditures, can impact the feasibility or strategy of CCM implementation [8]. Countries with publicly funded healthcare systems may have more resources to support implementation and reimbursement for integrated care services. Economic factors such as national health spending, workforce development policy, and investment in infrastructure may influence how a country creates and supports CCM initiatives [8]. Locations with fewer resources and rural settings may leverage telehealth to implement the CCM [13]. Because of the variation in health and government structures, patient diversity, and economic factors across the world, a systematic review is needed to provide a comprehensive understanding of the results of CCM implementation. The objectives of this review are as follows: (1) examine the effects of the CCM on symptom management in patients with GAD, MDD, or PTSD; (2) understand the implications of the CCM in the care of diverse populations; and (3) assess implementation strategies for and the longevity of the CCM in healthcare systems.

## 2. Materials and Methods

To understand the individual, population, and systemic impact of implementing the CCM, we reviewed the PubMed, PsychINFO, and Scopus databases to identify articles written in English from January 2008 to January 2024. Search terms relevant to the CCM included coordinated care, collaborative care, integrated care, interprofessional communication, interdisciplinary treatment, patient aligned care team, psychiatry, primary care, family medicine, behavioral health, mental health, and family healthcare. Search terms relevant to the conditions studied included depression, anxiety, and post-traumatic stress disorder. Our methods were published in PROSPERO on 22 December 2023 [ID CRD42024496198]. The systematic review was presented in compliance with the Preferred Reporting Items for Systematic Reviews and Meta-Analysis (PRISMA) guidelines from identification, screening, to final inclusion of eligible publications [14]. Final publications were evaluated for quality using the Let Evidence Guide Every New Decision (LEGEND) framework, a validated approach for assessing evidence for clinical decision making (see Appendix A) [15,16]. 

Inclusion criteria limited studies to those that (1) were published between January 2008 and January 2024; (2) were written in the English language; (3) analyzed adult patients; (4) analyzed the impact of CCM on either MDD, GAD, or PTSD; and (5) utilized CCM as defined by the triad of a primary care clinician, mental health provider, and patient care manager. Exclusion criteria excluded studies that (1) were published before 2008; (2) were published in a language other than English; and (3) were written as narrative or non-peer reviewed literature.

## 3. Results

### 3.1. Overview

The initial search resulted in 9743 articles, of which 119 were excluded for incomplete or missing key information (Figure 2). Examples of the studies also excluded for not meeting our criteria included those with the aim of qualitatively analyzing perspectives on the model rather than assessing quantified changes following CCM implementation, such as disease severity or cost [17,18]. Additionally, articles that provided only guidelines or expanded study protocols for CCM implementation were also excluded from the analysis [19,20]. A further 4583 duplicate articles were removed from consideration, preserving only one copy of each article. The titles and abstracts of the remaining articles were assessed using the above criteria, yielding 468 articles for an in-depth review. A full-text analysis of the remaining articles yielded 16 that fully met the inclusion and exclusion criteria. Figure 2 illustrates the flow of article extraction, after which comprehensive information from each selected article was recorded for analysis. While Figure 3 shows an increase over time in the number of articles related to the CCM, not all focused on the mental health conditions of adult patients. Of those meeting our search criteria, the number of articles varied by year (Figure 4), and no study relevant to health systems was published in the most recent three years.

### 3.2. Population Base, Fields of Authorship, and Targeting Outcomes

Of the 16 articles included for final analysis, 11 studies were conducted in the United States (US), 3 in the United Kingdom (UK), 1 in Spain, and 1 in Taiwan. Of the US studies, 2 were conducted in the southern region of the country versus 9 in the northern region. The sample sizes varied across the different levels of analysis. In the individual-level studies, the patient sample sizes ranged from 78 to 485. The population-level studies had sample sizes ranging from 60 to 5187. The systems-level analyses included clinic sample sizes between 3 and 459 and patient sample sizes from 387 to 103,000. The adult populations analyzed in this study included older adults (n = 485), women in general (n = 78), postpartum women (n = 78), Latina women (n = 60), veterans (n = 195), and rural patients (n = 5187). Rural-residing patients were defined as those served by a clinic located in a rural area labeled by the US federal government as “medically underserved” or having a “healthcare provider shortage” [21].

The 16 articles selected for the final analysis featured authors with backgrounds in family medicine, psychiatry, clinical psychology, environmental epidemiology, social work, health economics, primary care, bioethics and humanities, and nursing. The fields of publication for these articles included psychology, psychiatry, primary care, internal medicine, public health, healthcare implementation, and healthcare management. These articles were published across a variety of respected journals, notably, including the *British Journal of Psychiatry*, *the Journal of General Internal Medicine*, and *Primary Health Care Research & Development*.

To address our objectives, we analyzed our selected articles by category. The first category focused on articles that primarily examined the impact of the CCM on individual outcomes, specifically symptom management. An example of such an article can be seen in the work of Aragonès et al. [22], which detailed the long-term effects of the CCM on depression symptoms and severity. The second category included articles that examined the impact of the CCM on specific patient populations. A prime example of this is the article by Standeven et al. [23], detailing the effects of the CCM specifically on women. Finally, the third category included articles that explored the impact of the CCM on healthcare systems, specifically looking at implementation or sustainability. One such article is that by Moise et al. [24], which examined the sustainability of collaborative care in New York State-based clinics.

### 3.3. Measurement Tools

The improvement in the severity of individual symptoms was measured via screening tools, with five studies utilizing the Patient Health Questionnaire (PHQ-9) and two studies utilizing the Generalized Anxiety Disorder scale (GAD-7) surveys. The health outcomes in specified populations were measured using the Center for Epidemiological Studies-Depression Scale (CES-D), PHQ-9, and GAD-7 screening surveys. Several studies also examined suicidal ideation (SI) and the life satisfaction index (LSI). The PHQ-9 is a component of the longer Patient Health Questionnaire meant to offer providers with a concise tool for assessing depression based on Diagnostic and Statistical Manual of Mental Disorders-IV (DSM-IV) criteria [25]. The GAD-7 is a seven-item scale based on DSM-IV criteria for identifying likely cases of GAD [26]. CES-D is a measure of depressive experiences used in research and clinical practice [27].

The measurements to assess healthcare system effectiveness included quality-adjusted life years (QALYs), cost per member per month (PMPM), and the ability of clinics to sustain the CCM over time. The quality-adjusted life year, or QALY, is a measurement used in health economics that assesses the quality and quantity of life to determine the impact of an intervention. A value of 1 QALY represents one year of life in perfect health, 0.5 QALYs represents a reduction in the quality of life by 0.5, and 0 QALYs represents death [28]. Patients that receive treatment theoretically accumulate more QALYs in their lifetime and live longer than patients that do not. Next, the cost PMPM is another metric used in health economics to determine the monthly costs for each member covered by a particular health program. It is calculated as the annual cost of mental health services for each patient enrolled in the study, divided by 12 [29]. The ability of clinics to sustain the CCM over time was determined by the proportion of clinics that successfully continued with the model following a certain time period.

### 3.4. Individual Symptom Management

Examining the impact of the CCM on individuals within the US, four studies reported improvements in symptom management (Table 1). Sadock et al. found that patients from a predominantly low-income and ethnic minority background who received the CCM through an academic training clinic had a significantly greater decrease in depression (*p* = 0.003) and anxiety scores (*p* < 0.001) compared to those who received usual, unintegrated care [30]. The interval of follow-up did not differ significantly between the CCM and control clinic, with a median follow-up of 52–56 days from the initial mental health screening (*p* = 0.47) [30]. In 2017, Truitt et al. conducted a retrospective cohort study seeking to analyze the effects of the CCM on patients with postpartum depression enrolled from a multispecialty referral center who had an initial PHQ-9 score of 10 or greater. Those receiving the CCM had a significant improvement in the 6- and 12-month remission rates as well as fewer days to first follow-up (*p* < 0.01) [20]. No significant difference was seen in healthcare utilization between the CCM and routine care [31]. These findings suggest that the CCM can provide a higher level of care without an increase in the number or length of scheduled appointments. Sederer et al. found that, upon implementing the CCM across 32 different clinics in New York State, 46% of patients had PHQ-9 scores that fell below the clinically significant score of 10 after at least 16 weeks of follow-up [21]. Additionally, the clinics increased their frequency of providing depression screenings to all patients and, of those who received a positive depression screen, 66% received a diagnosis, compared to 44% prior to CCM implementation [21].

Out of the final articles analyzed, only one US-based study did not yield a significant improvement in symptom burden (Table 1). Based in Minnesota, Solberg et al. evaluated the effectiveness of the Depression Improvement Across Minnesota-Offering a New Direction (DIAMOND) initiative in implementing the CCM in 75 primary care clinics over a 6-month period [29]. Its implementation was staggered, with 10 to 40 clinics adopting the model every 6 months [29]. The overall rates of depressive symptoms did not differ significantly between the clinics employing the CCM and usual care (36.4% CCM vs. 33.9% usual care, *p* = 0.94), as evidenced by similar changes in the PHQ-9 scores [29].

The findings from Europe yielded less favorable results than those seen in clinics in the US. In older adults diagnosed with MDD, Bosanquet et al. found a significant improvement in depressive symptoms at 4 months (*p* < 0.001), but this improvement was not maintained upon long-term follow-up beyond 12 months [32]. However, the anxiety scores differed significantly at 4- and 12-month follow-up [32]. Aragones et al. analyzed the effectiveness of the CCM over a long-term follow-up period in patients diagnosed with MDD. They found that, while the PHQ-9 scores did decrease, the change did not significantly differ from that of those who received usual care at 3 years, possibly due to a small sample or to the long-term “low-intensity” intervention style of the study [22].

### 3.5. Population Impact

Examining the population-level impact of the CCM provides the additional nuance of its impact. While the CCM can improve mental health outcomes for many individuals, its success varies by populations. It appears to be particularly effective in treating populations who may be more vulnerable to developing mental illness as well as those with less access to mental health care. These vulnerable populations include women in general, Latinas in the US, veterans, people living in rural areas, and the elderly (Table 1) [39,40,41]. Several studies examined CCM implementation in these populations using the improvement in depression, anxiety, or PTSD as a primary metric for success. In addition, our studies also support better outcomes for patients participating in the CCM regardless of whether they belong to a potentially vulnerable population. 

Overall, the CCM interventions consistently reduced depression and anxiety symptoms significantly, as assessed by the PHQ-9 or CES-D and GAD-7, respectively. Specifically looking at outcomes related to depressive symptoms, Eghaneyan et al. examined the effect of CCM implementation at a federally qualified health center in Texas that primarily served low-income, Spanish-speaking women. A significant reduction in depressive symptoms from the baseline PHQ-9 scores was seen [33]. The mean baseline PHQ-9 score was 18.98 (standard deviation [SD] 5.35) and the mean final follow-up PHQ-9 score was 14.33 (SD 6.88) [33]. The researchers used a nonparametric, related-samples Wilcoxon signed-rank test to determine differences in depression scores, which yielded a test statistic of Z 3.82 (*p* = 0.001), suggesting a statistically significant decrease in the scores [33]. Powers et al. also found that CCM implementation in rural communities in the US led to a statistically and clinically significant improvement in the PHQ-9 scores and in SI [34]. In this study, the mean change in the PHQ-9 score from the first to last measurement was 5.1 points (SD 6.7; 95% confidence interval [CI] [4.9–5.3]), and the number of patients with SI decreased by 11%, with a mean decrease in SI severity of 0.25 on a 0–3 scale (*p* = 0.0001) [34]. Finally, Liao et al. demonstrated that CCM implementation in retirement communities in Taiwan led to statistically significant improvements in metrics associated with depression, including the CES-D, LSI, and SI [35]. Using the generalized estimating equation (GEE), they found that the CES-D, SI, and LSI scores steadily improved in the intervention group, after controlling for covariance [35]. Specifically, the GEE showed an interaction effect between groups (control vs. intervention) and time at 18 weeks (β = −4.76, 95%CI: −7.39 to −2.14, *p* < 0.001) in the CES-D score; at 12 weeks (β = 0.75, 95%CI: 0.45–1.05, *p* < 0.004) in SI; and at 12 weeks (β = 2.72, 95%CI: 0.48–4.97, *p* = 0.018) and at 18 weeks (β = 5.06, 95%CI: 2.57–7.54, *p* < 0.001) in the LSI [35]. Standeven et al. found a statistically significant reduction in anxiety, as assessed by GAD-7 scores, in US-residing women who received CCM care in association with women’s health centers [23]. When comparing the mean GAD-7 score at baseline (11.46, SD 5.09) to the final GAD-7 score (7.48, SD 5.18), a paired Student’s *t*-test revealed a significant improvement in the average anxiety symptoms [t (218) = 12.41, *p* < 0.001] [23].

While the above studies suggest that the CCM can be an effective tool for improving anxiety and depression outcomes in vulnerable patient populations, it did not prove as effective when treating Veteran Affairs (VA) patients with PTSD [36]. Schnurr et al. found that, when implementing the CCM at VA clinics in Texas, patients did see a significant improvement in PTSD symptoms (*p* < 0.001), but the improvement between the CCM intervention group and the usual care group was not significant [36]. In addition, the authors monitored depression symptoms and physical functioning. While these also improved over time (*p* < 0.001), the overall magnitude of change was small, and the two groups did not differ significantly [36].

### 3.6. Healthcare System Impact

Although the impact of the CCM on an individual and population basis was positive overall, it is essential to also examine its effect on healthcare systems, specifically in terms of healthcare costs and sustainability over time. In Minnesota, Angstman et al. conducted an observational study reviewing financial data from 2008 and 2009 from clinics that utilized either usual care or the CCM. The mental health cost PMPM was used as a metric of cost and included the services of emergency department visits, inpatient psychiatry stays, outpatient psychiatry consultation, and psychotherapy sessions [29]. In the 2-year period, the mental health costs associated with the CCM did not significantly differ from those associated with usual care [29]. However, the mental health PMPM significantly decreased in the CCM clinics from 2008 to 2009, from 4.91% to 4.36% (95%CI: 4.121–4.609%, *p* < 0.0001).

On the other side of the Atlantic, Camacho et al. analyzed 459 clinics in the northwest region of the UK and found an increase in healthcare costs associated with the CCM for patients with comorbid depression and diabetes or coronary heart disease. However, this increase was accompanied by a significant increase in QALYs over 24 months compared to usual care, with a cost of £13,069 per increase in QALY (0.136; 95%CI: 0.061–0.212) [28]. The probabilities of the CCM being cost-effective were 0.75 and 0.92 if payers were able to pay £20,000 and £30,000, respectively (the current 2024 exchange rate indicates that £20,000 is approximately $25,474) [28]. Also in the UK, Green et al. examined the cost-effectiveness of the CCM in clinics and found an increase of 0.02 QALYs (95%CI: 202.98–886.04) over a 12-month period with an estimated mean cost of £14,248 per QALY [37]. The improvement in depression symptoms and increased odds of depression recovery (1.67; 95%CI: 1.22–2.29) were achieved with the CCM with a non-significantly increased cost compared to usual care [28]. This finding was further supported by the cost-effectiveness analysis of Bosanquet et al., which reported that the average cost of delivering the CCM was £198 per patient compared to the initial £495 estimate [32]. Bosanquet et al. also noted an increase of 0.019 QALYs (*p* = 0.338) from baseline in the CCM participants, a compelling if not statistically significant finding [32].

In addition to healthcare costs, assessing the feasibility of the implementation and sustainability of the CCM over time is essential to understanding the model’s effectiveness. The factors that contribute to its successful implementation should be identified to ease the process in other regions. Bowen et al. analyzed the process implementation of the CCM in the New York State Collaborative Medicaid Program according to the Reach, Effectiveness, Adoption, Implementation, and Maintenance (RE-AIM) model [38]. They found that the state mental health program was able to reach 83% of the state’s eligible Federally Qualified Health Centers. An average of a 44.81% improvement in depression and anxiety symptoms was seen with no statistically significant difference between various training and technical assistance (TTA) sources or intensity (*p* = 0.144, *p* = 0.561, respectively), and 75% of the enrolled clinics utilized TTA. Ultimately 79% (n = 130) of the clinics were successful in maintaining the CCM for at least 1 year [38]. Lastly, Moise et al. examined factors that correlated with a clinic’s ability to sustain the CCM in the state of New York. Of the 32 primary care clinics followed, 26 clinics continued utilizing the CCM in a clinical setting [24]. The clinics that were able to sustain the CCM were more likely to demonstrate an improvement in clinical depression symptoms at a rate of 46.0% within the sustainability period vs. 7.5% in the opt-out system (*p* = 0.0004) [24]. The challenges to implementation included limited time and staff resources, provider and staff engagement and communication, and workflow logistics with complicated screening and referral processes [24]. 

## 4. Discussion

### 4.1. Overview

When considering the impact of our analysis, it is important to consider that we included publications from the three major components of the CCM: those from psychiatrists, care managers, and PCPs. Examining articles from major stakeholders allows us to have a more holistic understanding of the CCM and better supports our conclusions. Notably, we found that the CCM is effective in improving mental health outcomes in individual patients and in vulnerable populations [21,22,23,30,31,32,33,34,35,36,37,39,40,41]. Most of our final studies showed an improvement in patient symptoms, even in Europe, where the change was less dramatic [21,22,23,24,30,31,32,33,34,35,36,37,38,39,40,41,42,43]. The CCM appeared to be especially effective in reducing the self-reported symptom burden in female patients or those with anxiety or depression [21,23,30,31,32,33,34,35]. Furthermore, our analysis demonstrated sustainable models for the CCM, with New York State providing an example of successful implementation and longevity across several primary care clinics with sufficient governmental support [21,37,38]. We found that, with the appropriate allocation of funds, the addition of CCM Current Procedural Terminology (CPT) codes for Medicare, and sustained training and support, the CMM is both beneficial to patients and cost-effective [21,24,28,38].

Our findings offer new insight from previous reviews, as few studies have explored the perspective of three levels of impact. Yet, our findings do align with the conclusions of other review studies of the clinical and systemic impact of implementing the CCM. When considering symptom management, Muntingh et al. evaluated the impact of the CCM on adult patients with anxiety disorders through a systematic review and meta-analysis [44]. Of the seven articles analyzed, the authors found that the CCM significantly improved the management of anxiety disorders compared to in standard care [44]. A particularly large improvement was seen for those with panic disorder [44]. It was further noted that the impact on self-reported anxiety scale scores was greater in studies conducted in the US than in studies conducted in Europe [44]. The studies from Europe (I2 = 78%) were more heterogeneous than the studies from the US (I2 = 0%) and further research should be conducted to clarify this relationship [44]. These findings are similar to those from our study, as US patients with MDD had a greater improvement in symptom management than those in Europe [21,22,30,31,32].

Furthermore, our findings align with and build on the idea that the CCM can be used to deliver optimal care to specific populations. Huang et al. conducted a systematic review focusing primarily on women with depression in primary care and women’s health clinics [45]. Of the seven articles reviewed, six were randomized controlled trials and one was an observational study [45]. The authors found that, at an individual and population level, the CCM is an effective strategy to manage depression in women compared to usual care [45]. This result aligns with our findings, as women receiving management for MDD and GAD found a significant improvement when treated through the CCM [23,33]. Further studies may be needed to clarify the differences between identified genders in improvement through the CCM, or whether the significant impacts seen in female patients are unique to this patient population.

Finally, our findings expand on the discussion of the barriers to and solutions for the CCM. Ramanuj et al. analyzed the effectiveness of integrated behavioral care in the UK and US for patients with a broad range of conditions, such as serious mental illness and substance use disorders [46]. Along with the CCM, this review also included the Screening, Brief Intervention, Referral to Treatment (SBIRT) integrated care model, used primarily for alcohol and substance use disorders [46]. The primary emphasis was on overcoming challenges, such as policy and funding sources, to the implementation of integrated models within healthcare systems [46]. Our analyzed studies demonstrated that, if implemented on a large scale, a majority of clinics can maintain the CCM in the long term [24,38], with potentially no increase in cost compared to usual care, yet a significant increase in QALYs [28,29,32]. These findings indicate that, if funding sources can adapt appropriately, the CCM may provide significant improvements for patients with no increase in cost for healthcare systems. 

### 4.2. Recommendations for Policy, Practice, and Research

After our analysis, we suggest that Medicare and other payers consider updating their policies to include reimbursement for CCM care, given that its return on investment appears promising. However, given the current evidence, the broad implementation of the CCM on a nationwide scale may be premature without further studies conducted across a range of geographical regions and patient populations. Additionally, local-level allocation of public health funds for this effort, when possible, may be beneficial. Further, the results of this analysis suggest that best practices include providing comprehensive training to clinicians and care managers prior to CCM implementation [47]. Practices that wish to implement the CCM should first consider the populations which they will be serving, to determine if the CCM will be efficacious, then begin their efforts in clinics already serving the populations of interest. Once clinic sites have been identified, implementers should provide training, monitor outcomes through screening tools, and educate patients on the benefits of participating in the CCM [48]. The factors that have helped clinics implement the CCM include incorporating IT staff to assist with electronic medical record systems; using the CCM in telemedicine consultations; and providing all staff with technical training [24,48]. Finally, we recommend that future research on CCM efficacy continue to look at multiple levels of outcomes to identify which conditions see the best symptom improvement, what populations benefit most, and which implementation strategies are most cost-effective. 

### 4.3. Strengths

While several reviews have been conducted on the CCM, few have reported international outcomes and how they differ from findings in the US, where the model was first developed. Additionally, previous studies often analyze individual symptom management, population outcomes, or the impact on healthcare systems. Our study analyzes all three levels of outcomes across multiple diagnoses to explore the overall result of utilizing the CCM for both patients and healthcare systems, providing better context.

### 4.4. Limitations

Methodologically, only the three databases of PubMed, PsychINFO, and Scopus were searched and therefore this analysis may not encompass all relevant, published literature. The inclusion criteria of requiring the final literature to be written in English may also fail to include international studies that feature the CCM but are written in languages native to the country where the study was conducted. This may contribute to our final analysis being predominantly of literature from the US and the UK. Therefore, geographical limitations are present that may impact the generalizability of our findings to populations beyond the specific demographics and geographical regions studied. The only study based in Asia, by Liao et al., may be applicable to Asian elderly patients or elderly Asian American immigrants in the US living in retirement facilities with similar health profiles. The studies conducted outside the US often had healthcare systems that established the CCM on a large scale, so implementation may differ in places with more limited access to funding. In addition, the studies by Bowen et al. and Moise et al. were based out of New York State and may not be directly applicable to other regions within the US. Notably, the studies conducted within the southern and midwest regions of the US were limited in number. More studies must be conducted to fully understand the impact for patients in various populations and the sustainability in national healthcare systems of interest. All the final articles were analyzed through the Let Every Evidence Guide New Decision (LEGEND) system and were determined to be of good to moderate quality [15,16]. 

Lastly, an important consideration is that quality-adjusted life years (QALYs) were used as a measurement in two articles performing a systemic-level analysis based in the UK, Camacho et al. and Green et al. [28,38]. The Protecting Health Care for All Patients Act of 2023 (H.R. 485) prohibits the use of QALYs for deciding coverage and payment in US federal health programs such as Medicare and Medicaid [49]. The act aims to protect people with disabilities and chronic illnesses from discrimination that comes from health-related quality of life used to calculate QALYs. For this reason, we advise the use of alternative metrics in future studies to determine the impact of the CCM for the purpose of informing American healthcare policies. An option is the disability-adjusted life year (DALY), which is a measurement also commonly used in cost-effectiveness analysis that determines the overall burden of disease by combining the years of life lost due to premature death and the years lived with disability [50]. It provides a holistic perspective on health and is not likely to devalue the lives of individuals with disabilities [50].

## 5. Conclusions

The results of the analyzed studies demonstrate that the CCM can provide superior patient outcomes in studied adult populations compared to usual care when used to treat diagnosed depression and anxiety, notably reducing the reported symptom burden and improving remission rates. This impact was seen in populations vulnerable to developing mental illness or having limited access to care such as older adults, women, and adults without insurance, as seen by the evaluated studies. The CCM is additionally cost-effective when implemented well. A significant impact was not seen for managing PTSD in veteran populations; however, due to the limited number of studies focusing on PTSD, further studies may be needed to assess the true impact on this condition. The CCM can be implemented in existing primary care clinics to great effect, making it an exciting new method for delivering quality care to patients with limited access to care due to location or socioeconomic status, as well as the general adult population. The key aspects of the successful implementation and maintenance of the CCM noted by the analyzed studies included appropriate funding sources and adequate staff training. The included studies were assessed for quality through a LEGEND analysis and varied from good to moderate quality. The major findings across the studies agreed with each other, making us confident in our conclusions. Further studies are needed to assess the long-term monitoring of the longevity of the effects seen with the CCM globally and in the southern regions of the US, but current studies appear promising at both the national and international level.

## Figures and Tables

**Figure 1 healthcare-12-01679-f001:**
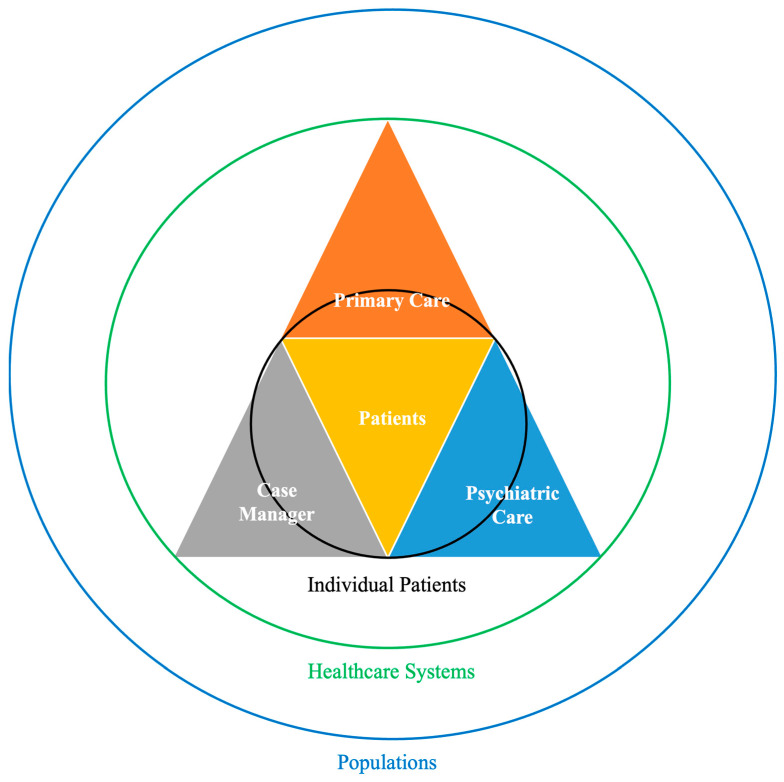
Visualization of the collaborative care model.

**Figure 2 healthcare-12-01679-f002:**
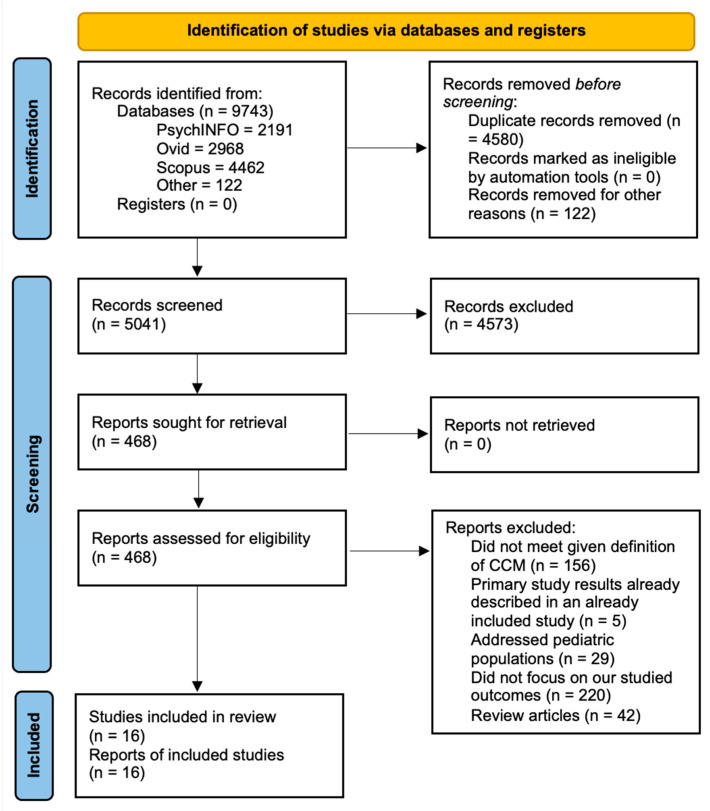
Systematic review flow diagram.

**Figure 3 healthcare-12-01679-f003:**
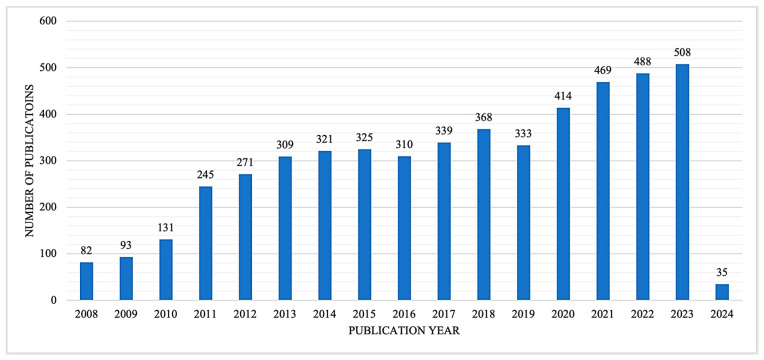
Number of articles included for screening after removal of duplicates and incomplete citations, by year.

**Figure 4 healthcare-12-01679-f004:**
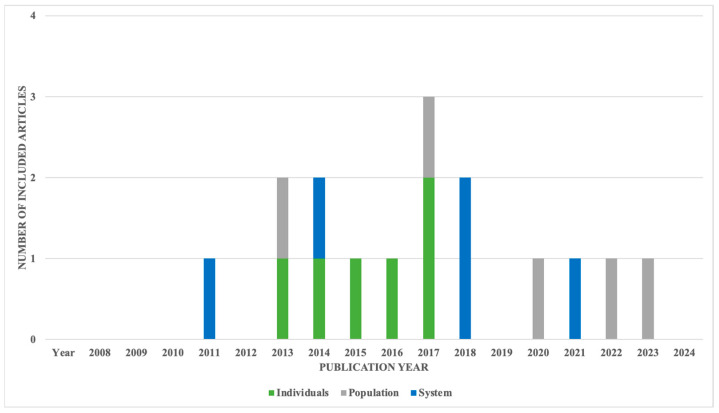
Articles included in final analysis, by year and category.

**Table 1 healthcare-12-01679-t001:** Results of systematic review: impacts of implementing CCM.

Author	Publication Year	Type of Article	Level(s) Analyzed		Country	Population	Condition(s) Assessed	Sample Size (Participants)	Sample Size (Clinical Sites)	Findings
Rossom, R.C. et al. [4]	2015	Quantitative	Individual		United States	Adults	MDD	1579	75	No significant change in PHQ-9 scores at 6-month follow-up
Camacho, E.M. et al. [28]	2018	Quantitative	System	Individual	United Kingdom	Adults	MDD	387	39	Improvement in SCL-D13 scores and QALYs at a cost of £13,069 per QALY
Angstman, K.B. et al. [29]	2011	Quantitative	System		United States	Adults	MDD	103,000	3	4.5% reduction in per member per month cost after 2 years of implementation
Sadock, E et al. [30]	2017	Quantitative	Individual		United States	Adults	MDD and GAD	286	2	Short- (6-month) and long-term (18-month) improvement in PHQ-9 and GAD-7 scores
Truitt, F.E. et al. [31]	2013	Quantitative	Individual	Population	United States	Postpartum women	PPD (subset of MDD)	78	2	Long-term improvement (12 month) in PHQ-9 scores and shorter time interval before first follow-up
Sederer, L.I. et al. [21]	2016	Mixed methods	Individual	System	United States	Adults	MDD	6000 (estimated)	32	Improvements in PHQ-9 scores < 10 when treated under CCM for at least 16 weeks
Bosanquet, K et al. [32]	2017	Mixed methods	Individual		United Kingdom	Older adults 65+ years of age	MDD	485	69	Short-term improvement in PHQ-9 scores not maintained at 12–16-month follow-up
Aragonès, E et al. [22]	2014	Quantitative	Individual		Spain	Adults	MDD	338	20	Improvements in PHQ-9 scores not maintained at 36-month follow-up
Eghaneyan, B.H. et al. [33]	2017	Quantitative	Population		United States	Uninsured Latina women	MDD	60	1	Improvements in PHQ-9 scores at 12-month follow-up
Powers, D.M. et al. [34]	2020	Quantitative	Population		United States	Rural, low-income	MDD	5187	8	Improvements in PHQ-9 scores at 18-month follow-up
Liao, S.-J. et al. [35]	2022	Quantitative	Population	Individual	Taiwan	Older adults 55+ years of age	MDD	143	1	Improvements in CES-D scores at 18-week follow-up
Standeven, L et al. [23]	2023	Quantitative	Population		United States	Women	GAD	219	1	Improvement in GAD-7 scores at 3-month follow-up
Schnurr, P.P. et al. [36]	2013	Quantitative	Population	Individual	United States	Veterans	PTSD	195	5	No significant change in symptom burden at 3- or 6-month follow-up
Green, C. et al. [37]	2014	Quantitative	System		United Kingdom	Adults	MDD	581	3	Mean increase of 0.02 QALYs over 12 months with a mean increase cost of £14,248 per QALY
Bowen, D.J. et al. [38]	2021	Quantitative	System	Population	United States	Medicaid insured and uninsured adults	MDD and GAD	Not defined	611	Improvement in PHQ-P and GAD-7 scores after at least 1 year of implementation. Availability of technical support associated with increased rates of CCM maintenance
Moise, N. et al. [24]	2018	Mixed methods	System		United States	Not defined	MDD	Not defined	32	Higher staffing of care managers was associated with the likelihood of maintaining CCM and greater improvement in PHQ-9 scores

MDD = major depressive disorder; GAD = generalized anxiety disorder; PTSD = post-traumatic stress disorder; PHQ-9 = Patient Health Questionnaire 9; GAD-7 = Generalized Anxiety Disorder Questionnaire; CES-D = Center for Epidemiological Studies Depression Scale; QALYs = Quality-adjusted life years; SCL-D13 = Symptom-Checklist Depression Score.

## Data Availability

No new data were created or analyzed in this study.

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
