# Peer review of "Healing the Whole: An International Review of the Collaborative Care Model between Primary Care and Psychiatry"

_healthcare, 2024, doi:10.3390/healthcare12161679_

Round 1

Reviewer 1 Report

Comments and Suggestions for Authors

Healing the whole: An international Review.

Topic

The topic has substantial interest for the public and other researchers.

Title

The title is an International Review. It seems that International can be deleted to emphasize that this is not a systematic review. 

Aim of paper

Is to understand the impact of CCM on symptom management, diverse populations, and sustainability in health care systems.

Structure of paper

The paper follows the IMRAD structure and includes 4 figures and no tables.

Methodology

The paper is registered in Prospero which accepts; systematic reviews, rapid reviewsand umbrella reviews. It is not stated which type of review this paper undertakes, which makes it difficult to review the paper properly, however, it primarily resembles a literature review.

In the paper the authors state that: “The literature review was presented in compliance with the Preferred 88 Reporting Items for Systematic Reviews and Meta-Analysis (PRISMA) guidelines from 89 identification, screening, to final inclusion of eligible publications”. There is no reference to this statement. The latest update of the PRISMA guideline is from 2020.

https://www.prisma-statement.org

In the Prospero registration, the author states that they will use the ROB 2 to assess the quality of the included papers, however, this is not accounted for in the paper. It will make it easier to assess the quality of the paper  and the evaluate the basis of the conclusion if some of the methodology mentioned in PROSPERO is also included in the study.

The search strategy would benefit if studies that might appear to meet the inclusion criteria, but which were excluded are cited and an explanation is put forward why they were excluded.

Figure 2 will benefit from following the PRISMA recommendation for flow diagrams: https://www.eshackathon.org/software/PRISMA2020.html.

Even though this is not a systematic review the paper would greatly benefit from adhering to some parts of the PRISMA checklist 2020.

Results

No tables are presenting the methodology and results of the individual papers, nor is there any assessment of quality. It will enhance the readability of the paper if the methodology, quality assessment (ROB 2), and results of the original papers are presented in tables.

Discussion

The limitation section of the paper does not contain any discussion of the methodological limitations of the review, as the section primarily touches upon the generalisability of the results.

Conclusion

The paper concludes that CCM demonstrated to provide the same or better patient outcomes as usual care.

However, the lack of quality assessment of the original papers makes it impossible to assess whether this rather definitive conclusion is warranted.

https://bestpractice.bmj.com/info/us/toolkit/learn-ebm/what-is-grade/

Overall conclusion:

It has been difficult to review the paper, as the review type is not properly defined by the authors.

The paper needs major revisions such as:

The type of review must be defined explicitly

The methodology must be expanded with some of the issues mentioned in PROSPERO.

The methodology, results, and quality assessment must also be presented in tables

The limitation section (in discussion) must include a discussion of the methodology of the quality assessment of the review as a whole and of the original papers

The conclusion must reflect the quality of the included papers

Author Response

Topic

The topic has substantial interest for the public and other researchers.

  • Thank you for your valuable feedback.

Title

The title is an International Review. It seems that International can be deleted to emphasize that this is not a systematic review.

  • The title is unchanged due to the manuscript being a systemic review.

Aim of paper

Is to understand the impact of CCM on symptom management, diverse populations, and sustainability in health care systems.

Structure of paper

The paper follows the IMRAD structure and includes 4 figures and no tables.

Methodology

The paper is registered in Prospero which accepts; systematic reviews, rapid reviews, and umbrella reviews. It is not stated which type of review this paper undertakes, which makes it difficult to review the paper properly, however, it primarily resembles a literature review.

  • The type of review is stated in line 88 of the materials and methods section.

In the paper the authors state that: “The literature review was presented in compliance with the Preferred 88 Reporting Items for Systematic Reviews and Meta-Analysis (PRISMA) guidelines from 89 identification, screening, to final inclusion of eligible publications”. There is no reference to this statement. The latest update of the PRISMA guideline is from 2020. https://www.prisma-statement.org

  • PRISMA guidelines citation was added to line 90.

In the Prospero registration, the author states that they will use the ROB 2 to assess the quality of the included papers, however, this is not accounted for in the paper.

  • All articles included in the study were re-assessed for quality through LEGEND analysis, lines 91-92. LEGEND framework was added in Appendix 1.

It will make it easier to assess the quality of the paper and evaluate the basis of the conclusion if some of the methodology mentioned in PROSPERO is also included in the study.

  • We revised the introduction with our study’s objectives (Lines 73-78) and addressed the objectives in the conclusions (lines 323-333, 348-357).

The search strategy would benefit if studies that might appear to meet the inclusion criteria, but which were excluded are cited and an explanation is put forward why they were excluded.

  • The examples of excluded studies were stated in lines 104-108 of the overview sub-section of results.

Figure 2 will benefit from following the PRISMA recommendation for flow diagrams: https://www.eshackathon.org/software/PRISMA2020.html.

  • We added a section for the reports excluded in Figure 2.

Even though this is not a systematic review the paper would greatly benefit from adhering to some parts of the PRISMA checklist 2020.

  • The aspects of the PRISMA 2020 checklist were expanded on throughout the paper.

Results

No tables are presenting the methodology and results of the individual papers, nor is there any assessment of quality. It will enhance the readability of the paper if the methodology, quality assessment (ROB 2), and results of the original papers are presented in tables.

  • We added Table 1 that displays the methodology and results of the papers included in the study.

Discussion

The limitation section of the paper does not contain any discussion of the methodological limitations of the review, as the section primarily touches upon the generalizability of the results.

  • We added discussion about the methodology to the limitations section, specifically study design, language, and quality analysis in lines 398-405

Conclusion

The paper concludes that CCM demonstrated to provide the same or better patient outcomes as usual care.

However, the lack of quality assessment of the original papers makes it impossible to assess whether this rather definitive conclusion is warranted.

https://bestpractice.bmj.com/info/us/toolkit/learn-ebm/what-is-grade/

  • The quality assessment was stated in lines 91, 415-416, and 445.

Overall conclusion:

It has been difficult to review the paper, as the review type is not properly defined by the authors.

  • The type of review was stated in line 88 of the materials and methods section.

The paper needs major revisions such as:

The type of review must be defined explicitly

  • The type of review was stated in line 88 of the materials and methods section.

The methodology must be expanded with some of the issues mentioned in PROSPERO.

  • Our conclusion addresses RQ1 in lines 323-328, RQ2 in lines 328-333 and 384-389, and RQ3 in lines 335 and 409-411.

The methodology, results, and quality assessment must also be presented in tables

  • We expanded Table 1.

The limitation section (in discussion) must include a discussion of the methodology of the quality assessment of the review as a whole and of the original papers

  • We expanded the study limitations in lines 398-405.

The conclusion must reflect the quality of the included papers

  • We added the quality statement in line 445.

Reviewer 2 Report

Comments and Suggestions for Authors

The authors must eradicate plagiarism, and use their phrases if they can and could. At the same time, they must extend now or in further investigations, the applicability to other countries and languages, not only in Europe. They also try to implement the studied model outside the limited time and personnel resources to improve communication and dynamization of the work, extend the focus on other mental health conditions, provide more details about the selection process for the study, and provide more detailed discussion implications. Congratulations on improving the work.

Comments on the Quality of English Language

The quality of English is acceptable only improving consistency in some syntax aspects, only.

Author Response

The authors must eradicate plagiarism and use their phrases if they can and could. At the same time, they must extend now or in further investigations, the applicability to other countries and languages, not only in Europe. They also try to implement the studied model outside the limited time and personnel resources to improve communication and dynamization of the work, extend the focus on other mental health conditions, provide more details about the selection process for the study, and provide more detailed discussion implications. Congratulations on improving the work.

  • Thank you for your valuable feedback. We have properly cited content and rephrase the content to avoid plagiarism.

Reviewer 3 Report

Comments and Suggestions for Authors

This paper addresses a topic of interest, namely the impact of the implementation of the Collaborative Care Model.

The structure of this research is coherent, logical and orderly.

The bibliography is recent and well cited, with no citation errors detected.

The only part that could be improved are the conclusions. In fact, they cannot really be called conclusions, as they are devoted to talking about the benefits of the CCM instead of summarising the findings established from the objectives of the review. Moreover, in any case, the conclusions are too brief.

It would also be appropriate to quote the objectives of the review at the end of the introduction, which could be answered in the conclusions.

Author Response

This paper addresses a topic of interest, namely the impact of the implementation of the Collaborative Care Model.

The structure of this research is coherent, logical and orderly.

The bibliography is recent and well cited, with no citation errors detected.

The only part that could be improved are the conclusions. In fact, they cannot really be called conclusions, as they are devoted to talking about the benefits of the CCM instead of summarizing the findings established from the objectives of the review. Moreover, in any case, the conclusions are too brief.

  • Thank you for your valuable feedback. We have strengthened our Conclusion section.

It would also be appropriate to quote the objectives of the review at the end of the introduction, which could be answered in the conclusions.

  • We revised the introduction with our study’s objectives (Lines 73-78) and addressed the objectives in the conclusions (lines 323-333, 348-357).

Reviewer 4 Report

Comments and Suggestions for Authors

Regarding the scope of the author’s findings:

The literature review shows benefits of CCM on MDD/GAD vs standard of care, but the reported findings are insufficient in my opinion to change practice standards/influence nationwide policy as the authors suggest. Authors seem to make a broad recommendation to “implement and reimburse for CCM” on a nationwide scale:

⁃ Authors never clarify if CCM is effective in “non-vulnerable” patient populations

⁃ Authors state that “NY State is an excellent proof of concept”, but what about other states? More rural regions in Midwest?

I do think the study includes enough data to say something along the lines of “CCM is beneficial and should be implemented for populations of interest in particular regions” but I think at present, it would be overstating the findings to suggest nation-wide change at this time.

Technical issues with the report:

Biggest issue involves demographic data, the authors are ambiguous in terms of what population size/demographics support each point. How many patients are included in analyzing impacts of CCM on individual symptoms vs population outcomes vs impact on healthcare systems?

⁃ “Patient sample sizes ranged from 25 to 5187. For systemic level analysis, clinic sample 122 sizes varied from 3 to 459 clinics. The adult populations analyzed included older adults, 123 women in general, postpartum women, Latina women, African American adults, veter-124 ans, and rural patients.” Line 122-124: this is all that is really offered by the authors throughout the article.

⁃ i.e., they mention a variety of “vulnerable” populations (women, low SES, rural). How many of each were included? What meets defintion or rural/low SES

⁃ Authors use a single study involving older adults in Taiwan and seem to use it to support implementation of CCM in the US - I do not think this is wise

I would like to see a citation for QALYs/PMPM. The authors should show that these metrics have been used elsewhere or cite a study indicating they are valid for target outcomes

Strong points of the report

Introduction:

I do like the comparison of CCM to other models of integrated psychiatric and primary care and think it introduces the concept well

Materials/Methods:

Solid, allows for reproducibility, using PRISMA guidelines and being stating the protocol was registered with PROSPERO increases their credibility

Figure 2: The flow chart is nice

Author Response

The literature review shows benefits of CCM on MDD/GAD vs standard of care, but the reported findings are insufficient in my opinion to change practice standards/influence nationwide policy as the authors suggest. Authors seem to make a broad recommendation to “implement and reimburse for CCM” on a nationwide scale:

  • Thank you for your valuable feedback.
  • We adjusted the recommendations section to suggest the consideration of policies for CCM but recognized that broad implementation would be premature with the current evidence. (Lines 372-383).

⁃ Authors never clarify if CCM is effective in “non-vulnerable” patient populations

  • In the conclusion section, we clarified that CCM can be utilized in primary care settings for both vulnerable and non-vulnerable populations. (Lines 439-441).

⁃ Authors state that “NY State is an excellent proof of concept”, but what about other states? More rural regions in Midwest?

  • We revised this sentence (Lines 410-414) to clarify that while New York State serves as an example of implementation and sustainability, it is not the sole possible example.

I do think the study includes enough data to say something along the lines of “CCM is beneficial and should be implemented for populations of interest in particular regions” but I think at present, it would be overstating the findings to suggest nation-wide change at this time.

  • We agreed with your concern. Multiple pilot projects in local levels were supported but not sustained to present the long-term value that we want to study.

Technical issues with the report:

Biggest issue involves demographic data, the authors are ambiguous in terms of what population size/demographics support each point. How many patients are included in analyzing impacts of CCM on individual symptoms vs population outcomes vs impact on healthcare systems?

  • The patient sample size ranges were specified for the individual, population, and system analysis (lines 134-139).

⁃ “Patient sample sizes ranged from 25 to 5187. For systemic level analysis, clinic sample 122 sizes varied from 3 to 459 clinics. The adult populations analyzed included older adults, 123 women in general, postpartum women, Latina women, African American adults, veter-124 ans, and rural patients.” Line 122-124: this is all that is really offered by the authors throughout the article.

  • This systematic review sought to identify comparable studies in each of three domains: individual, system, and population levels of outcomes. Certain studies with smaller sample sizes are qualified as they provide “individual” or “system” outcomes.

⁃ i.e., they mention a variety of “vulnerable” populations (women, low SES, rural). How many of each were included? What meets definition or rural/low SES

  • The number of each vulnerable population included in the study were specified in Lines 134-139. The definition of rural was added to Line 141.

⁃ Authors use a single study involving older adults in Taiwan and seem to use it to support implementation of CCM in the US - I do not think this is wise

  • The Taiwan study was discussed in the limitations section as an example of how study findings in one geographical area may not be generalizable to the US. (Line 405-407).

I would like to see a citation for QALYs/PMPM. The authors should show that these metrics have been used elsewhere or cite a study indicating they are valid for target outcomes

  • Citations for QALYs (Line 421) and PMPM (Line 183) were added. Limitations of using QALYs for calculations of federal coverage were also added (Line 428).

Round 2

Reviewer 4 Report

Comments and Suggestions for Authors

All the suggestions were well addressed .